# Intra- and Inter-Rater Reliability, Parallel Test Reliability, and Internal Consistency of the Tuning Fork and Monofilament Tests

**DOI:** 10.3390/bioengineering12111265

**Published:** 2025-11-18

**Authors:** Jitka Veldema, Lea Sasse, Jan Straub, Michel Klemm, Leon von Grönheim, Teni Steingräber

**Affiliations:** Faculty of Psychology and Sports Science, Bielefeld University, 33615 Bielefeld, Germany; lea.sasse@uni-bielefeld.de (L.S.); jan.straub@uni-bielefeld.de (J.S.); michel.klemm@uni-bielefeld.de (M.K.); leonvongroenheim@gmx.de (L.v.G.); teni.steingraeber@uni-bielefeld.de (T.S.)

**Keywords:** somatosensation, tuning fork test, monofilament test, intra-rater reliability, inter-rater reliability, parallel test reliability, internal consistency, young healthy adults

## Abstract

**Objectives:** Somatosensation is the ability to detect various external and internal stimuli (such as pain, pressure, temperature, or joint position), and its objective and reproducible evaluation is essential for diagnosis, training, and rehabilitation. This study evaluates the methodological quality of two somatosensory assessments in young healthy adults. **Methods:** The tuning fork test (administered on five locations of each hemibody) and the monofilament test (administered on 27 locations of each hemibody, and divided into (i) foot and ankle, (ii) leg and thigh, and (iii) trunk subscales) were applied to 58 students by two raters at three different time points (rater 1 test, rater 1 retest, rater 2 test). The intra- and inter-rater reliability, parallel test reliability, and internal consistency were evaluated for each test and subtest. **Results:** The tuning fork test showed moderate intra- and inter-rater reliability and good internal consistency. The monofilament test showed good to moderate intra- and inter-rater reliability for foot and ankle locations, but poor intra- and inter-rater reliability for leg, thigh, and trunk locations. The total score, left hemibody score, and right hemibody score of the monofilament test showed good or acceptable consistency with leg and thigh subscales, but poor or unacceptable consistency with foot, ankle, and trunk subscales. No acceptable parallel test reliabilities were found between the tuning fork test and the monofilament test. **Conclusions:** The tuning fork test is a reliable assessment of deep somatosensory function in the lower extremities of healthy young adults. The commercially available monofilament test kits are sufficient to investigate the superficial somatosensitivity of feet and ankles, but are insufficient for an objective evaluation of leg, thigh, and trunk regions.

## 1. Introduction

Somatosensation, defined as the ability to detect, interpret, and respond to sensory inputs (such as touch, pain, temperature, vibration, or changes in body position), plays a critical role in many areas of life. Several studies indicate that superior somatosensory control is associated with better outcomes in daily life, sports, rehabilitation, and prevention [1,2,3,4,5,6]. For example, gymnasts demonstrate superior proprioception in their upper limbs compared to untrained individuals [1]. Poor balance and frequent falls in individuals with unilateral transtibial (below-the-knee) amputation are correlated with reduced accuracy in touch and vibration detection of their lower extremities [2]. Healthy individuals typically exhibit a gradual decline in balance and gait control after the age of twenty, accompanied by worsened touch and vibration detection in their lower limbs [3]. Similarly, the thermal sensitivity to both warm and cold stimuli decreases continuously between 18 and 90 years of age [4]. Regular Tai Chi exercises in older adults may support their balance control, as well as proprioception of their lower limbs [5]. Regular sensory foot exercises in patients with chronic ankle instability improve balance control and ankle stability, together with improved proprioception and vibration sensing within lower limbs [6]. Thus, an objective evaluation of somatosensation is of great interest for both research and practice. Nonetheless, the existing evaluation tools still leave much to be desired.

In general, one differentiates between superficial and deep somatosensitivity [7]. Superficial sensation includes touch, pain, temperature, and two-point discrimination [7]. Deep sensation includes muscle and joint position sense (proprioception), deep muscle pain, and vibration sense [7]. Several instruments have been developed, such as the Rydel-Seiffer tuning fork [8], electronic vibrameters [8], proprioception test apparatus [9], two-point discriminators [10], and Semmes–Weinstein monofilaments [11], to enable the investigation of somatosensitivity types in their full range and diversity. However, their application in research and practice is inconsistent, and no comprehensive, standardised test batteries exist, as are commonly available in other research areas. Examples include the Berg Balance Scale [12], the Wolf Motor Function Test [13], and the 36-Item Short Form Health Survey [14], which are established, reliable assessments to evaluate balance control [12], hand motor function [13], and health-related quality of life [14] using defined scores and items. Our study aims to support the development of objective measures in the field of somatosensory control and to evaluate the methodological quality of two frequently used instruments: the Semmes–Weinstein monofilaments and the Rydel–Seiffer tuning fork.

The Semmes–Weinstein monofilament test evaluates superficial touch sensitivity by applying monofilaments with different thicknesses over the area of interest. The number and thickness of the monofilaments, as well as the site of their application, vary widely across present research [3,15,16]. For example, twenty monofilaments (thickness 1.65 to 6.65) across six regions (first distal phalanx, first and fifth metatarsal head, heel, instep, first interosseal space) were applied to test foot sensation in healthy people of different ages [3]. Seventeen monofilaments (thickness 4.20 to 6.30) were applied to the plantar surface of the hallux to evaluate foot sensitivity in a cohort of young healthy adults [15]. A single monofilament (5.07) on the big toe dorsum tested superficial foot sensation in healthy older adults [16]. The methodological quality of so many different approaches is often only insufficiently documented. Our study aims to address this gap in evidence by testing the intra- and inter-rater reliability and the internal consistency for 27 different foot, leg, thigh, and trunk regions using 20 monofilaments.

The Rydel–Seiffer tuning fork test evaluates deep vibration sensitivity by applying a 128 Hz vibrating tuning fork (with slowly decreasing oscillation amplitude) against a bony prominence. Existing applications show inhomogeneity regarding targeted areas [2,6,17]. One study applied a tuning fork over four regions (medial and lateral malleolus, tibial tuberosity, and patellae) to assess vibration sensitivity of lower limbs in patients with unilateral transtibial amputation [2]. Another study targeted three regions (tibial tuberosity, medial malleolus, and first metatarsal head) to test deep sensitivity of lower legs in people with chronic ankle instability [6]. Only the hallux was tested with a tuning fork to determine the vibration sensitivity of the lower limbs in patients with diabetes [17]. There is only limited evidence regarding the reliability of all these approaches. Our study will test the intra- and the inter-rater reliability and the internal consistency for five different lower-extremity areas and perform additional parallel-test comparisons with the Semmes–Weinstein monofilaments test.

## 2. Methods

### 2.1. Study Design

In this observational repeated-measures reliability study, two different assessments (tuning fork test and monofilament test) were applied by two raters (rater 1 and rater 2) at three time points (rater 1 test, rater 1 retest, and rater 2 test) in a randomised order, with at least 48 h between sessions. The participants were lying face up on a therapy table, and the somatosensitivity of both the right and the left hemibodies was tested in a randomised order. After the data were gathered, test–retest reliability, inter-rater reliability, parallel test reliability, and internal consistency were computed. The study was conducted in accordance with the Declaration of Helsinki and was approved by the Ethics Committee of Bielefeld University (EUB-2023-080).

### 2.2. Participants

A total of 58 healthy students were included in this study (age 24.4 ± 2.2 years; 28 females, 30 males; 47 right-footed; 11 left-footed). The foot preferred for kicking a ball was considered dominant [18]. A calculation with a power of 80% and a significance level of 0.05, adjusting for multiple testing (15 *) and assuming a true correlation of 0.5, indicates that a sample size of 59 is required.

### 2.3. Somatosensation Tests

#### 2.3.1. Tuning Fork Test

A Rydel–Seiffer tuning fork (Kirchner and Wilhelm GmbH + Co. KG, Asperg, Germany) with a scale of 0 to 8 (higher scores indicating better vibration sensitivity) was used to evaluate the vibration sensation of the (a) first metatarsophalangeal joint, (b) malleolus medialis, (c) malleolus lateralis, (d) patella, and (e) anterior superior iliac spine (Figure 1a). Three attempts were conducted at each location, and the mean values were used in the analyses [19].

#### 2.3.2. Monofilament Test

Twenty Semmes–Weinstein monofilaments with thicknesses ranging from 1.65 to 6.65 (smaller thickness indicates better pressure sensitivity) were used (Fabrication Enterprises Inc., New York, NY, USA). The pressure sensitivity was tested on 27 locations, divided into three subscales: foot and ankle (points 1–11), legs and thighs (points 12–21), and trunk (points 22–27) subscales (Figure 1b). Each location was touched three times consecutively with each monofilament, in a descending order of thickness. The smallest correctly detected filament diameters (at least two correct identifications) were used for the analyses [20].

### 2.4. Statistical Analysis

The SPSS software package, version 27 (International Business Machines Corporation Systems, IBM, Ehningen, BW, Germany) was used to analyse the data collected in this study. Intraclass correlation coefficients (ICCs) were used to evaluate intra-rater reliability, inter-rater reliability, and parallel test reliability (ICC ≥ 0.9 = excellent; 0.9 > ICC ≥ 0.75 = good; 0.75 > ICC ≥ 0.5 = moderate; and ICC < 0.5 = poor) [21,22]. Cronbach’s α coefficients were used to assess internal consistency (α ≥ 0.9 = excellent; 0.9 > α ≥ 0.8 = good; 0.8 > α ≥ 0.7 = acceptable; 0.7 > α ≥ 0.6 = questionable; 0.6 > α ≥ 0.5 = poor; α < 0.5 = unacceptable) [22,23]. The potential non-linear relationships were checked using scatter plots.

## 3. Results

An overview of the data (means and standard deviations) collected during the experiment is presented in Table 1 (tuning fork test) and Table 2 (monofilament test).

### 3.1. Intra-Rater Reliability

The tuning fork test showed moderate test–retest reliability for both the main score and its items (Table 1). The monofilament test demonstrated good or moderate test–retest reliability for the foot and ankle (points 1–11), and poor test–retest reliability for the legs and thighs (points 12–21) and trunk (points 22–27) subscales (Table 2).

### 3.2. Inter-Rater Reliability

The tuning fork test demonstrated moderate inter-rater reliability for the main score and for the major part of its items (Table 1). The monofilament test demonstrated good to moderate inter-rater reliability for foot and ankle, but poor inter-rater reliability for (i) the legs and thighs and (ii) trunk subscales (Table 2).

### 3.3. Internal Consistency

The tuning fork test and its items demonstrated good internal consistency (Table 3). The monofilament test overall scores (total, left hemibody, and right hemibody) demonstrated good or acceptable internal consistency in alignment with the leg and thigh subscale, but poor or unacceptable consistency with both the (i) foot and ankle and (ii) trunk subscales (Table 4). The internal consistency of the foot and ankle subscale (points 1–11) varied considerably (between excellent and unacceptable) (Table 5). Points 4–8 demonstrated the highest, while points 9–11 exhibited the lowest levels of internal consistency. The leg and thigh subscale (points 12–21) showed excellent to questionable consistency only for a small number of items, while a majority of the items demonstrated poor or unacceptable internal consistency (Table 6). Adjacent items (for example, points 12, 13, 14, or points 19, 20, 21) and items on the same hemibody were associated with significantly stronger internal consistencies than distant items (for example, points 12 and 21) and items on the opposite hemibody. The trunk subscale (points 22–27) demonstrated questionable to excellent consistency for the majority of the items (Table 7). Adjacent items on the same hemibody showed stronger consistencies than distant items and items on the opposite hemibody.

### 3.4. Parallel Test Reliability

The tuning fork test and the monofilament test, as well as all items/subscales, demonstrated unacceptable parallel test reliability (Table 8).

## 4. Discussion

This study investigated the methodological quality of two somatosensory assessments, the tuning fork test (applied on five points of each hemibody) and the monofilament test (applied on 27 points of each hemibody and divided into (i) foot and ankle, (ii) leg and thigh, and (iii) trunk subscales). The main findings were as follows: (i) the tuning fork test and its items demonstrated moderate intra- and inter-rater reliability and good consistency; (ii) the monofilament test demonstrated good or moderate intra- and inter-rater reliability only for the foot and ankle subscale and at least questionable internal consistency only for a portion of items and subscales; (iii) there was no relevant parallel test reliability between the tuning fork test and the monofilament test, as well as their items and/or subscales. Non-linear relationships have not been found.

### 4.1. Tuning Fork Test

Only a few studies have investigated the methodological quality of the tuning fork test in the field of somatosensory control [17,24,25,26]. The majority of the present data indicate satisfactory inter- and intra-rater reliability, consistent with our results [17,24,25,26]. For example, examinations on the wrist showed moderate inter-rater variability in healthy young (25 ± 5 years) participants [27]. The applications of the tuning fork test over eight limb and thigh locations showed good intra- and inter-observer agreement in patients with polyneuropathy [24]. The conventional, graduated, and dampened tuning fork tests over the hallux showed moderate intra- and substantial inter-rater reliability in patients with diabetes mellitus [17]. Poor to moderate inter-rater reliability was detected for applications over five bony sites on both feet in a diabetes mellitus cohort [25]. Even though a wide range of regions was investigated in existing studies, internal consistencies were not assessed, and our study addresses this gap in the evidence. Our data show that, despite good to excellent consistencies between individual items, the overall scores (total, right hemibody, and left hemibody) do not show significant relationships with them. This should be considered in both research and practice-oriented applications of the tuning fork test. The consistencies of the tuning fork test with other somatosensory assessments have been investigated in only two studies to date [25,27]. The data demonstrate that the tuning fork test and technically sophisticated tests of vibration sensation deliver consistent results [25,27]. Measurements with a vibrometer and tuning fork applied over the hallux showed moderate to high correlations in healthy young (25 ± 5 years) participants [27]. Strong correlations were reported between measurements obtained using a neurothesiometer and a tuning fork applied at five sites on the feet in patients with diabetes mellitus [25]. Comparisons with other somatosensory domains, such as touch sensation, two-point discrimination, or body position and/or movement sensing, have not been previously assessed, and our study provides the first evidence in this field. Our data indicate that no significant relationships exist between superficial touch sensation (tested by monofilaments) and vibration sensing (tested by tuning fork).

### 4.2. Monofilament Test

To the best of our knowledge, no previous study has tested the methodological quality of the monofilament test in as comprehensive a manner as our study. The major part of existing experiments in both healthy and disabled cohorts focused on foot areas only and confirmed satisfactory intra- and inter-rater reliability, consistent with our findings [15,16,17,21,28,29]. For example, seventeen monofilaments (4.20 to 6.30) applied to plantar hallux showed excellent intra-rater reliability in a young (23 ± 3 years) healthy cohort [15]. Investigations with a single monofilament (5.07) over the big toe dorsum showed excellent inter-rater reliability in community-dwelling older adults (50–89 years) [16]. A single monofilament (5.07) test over seven plantar and two dorsal foot areas demonstrated good intra- and inter-rater reliability in an older (61 ± 9 years) mixed cohort (healthy and diabetes/neuropathy) [26]. The application of the single monofilament (5.07) over four plantar foot areas showed fair to substantial intra- and substantial inter-rater reliability in diabetes mellitus patients [17]. The investigation of nine plantar and one dorsal foot area demonstrated moderate to substantial intra- and moderate inter-rater reliability in the same cohort [17]. Only a small portion of existing data contradicts the sufficient reliability of the Semmes-Weinstein monofilament for foot applications [20,28]. For example, applications of seven monofilaments (1.65 to 6.65) on five plantar foot areas showed not only good reliability for the left foot, but also moderate to poor intra-rater reliability for the right foot, as well as moderate to poor inter-rater reliability in a middle-aged (40 ± 11 years) healthy cohort [20]. Similarly, investigations with six monofilaments (2.83 to 6.65) over six plantar and two dorsal foot areas showed good to poor intra- and inter-rater reliability in diabetes patients (66 ± 13 years) [28]. There exist only a few studies that tested the reliability of Semmes–Weinstein monofilaments outside of foot areas [29,30]. Their results demonstrated good reliability for the hand area [29], but not for the shoulder, trunk, and lower limbs [30], consistent with our findings. For example, applications of twenty monofilaments (1.65 to 6.65) to the tips of the thumb and index finger showed good intra- and inter-rater reliability in stroke patients [29]. In contrast, twenty monofilaments (1.65 to 6.65) tested on four areas of the shoulder, abdomen, and lower limbs showed poor to unacceptable inter-rater reliability in healthy adults (21–68 years) [30]. It is plausible that larger body parts with few anatomical landmarks (such as legs or trunk) allow less precise (re)determination of the stimulation point, compared with smaller body parts with several anatomical landmarks (such as foot or hand). It is also evident that the commercially available Semmes–Weinstein monofilament kits are insufficient for an objective evaluation of trunk somatosensitivity in healthy people, as demonstrated by our data (Table 2). The addition of even thinner monofilaments could allow examination of the trunk region without ceiling effects. Despite the large body of evidence (15-17,20,26,30,31), hardly any data exist regarding the internal consistency of monofilament tests. Our study provides the first evidence in this field. Generally, good consistencies were observed between (i) the total score, (ii) the left and (iii) the right hemi-body scores, and (iv) the leg and thigh subscales. However, although both the trunk and foot–ankle subscales exhibited better internal consistency than the leg and thigh subscales, their consistency remained variable. Regarding parallel test reliability, little evidence exists so far [29,30,31]. Two studies showed that the monofilament test corresponds with other somatosensory assessments in disabled cohorts [29,31]. The thumb localising test correlated significantly with the monofilament test in a stroke cohort [29]. Similarly, both vibration perception threshold of the hallux (determined by biothesiometer) and the Ipswich Touch Test (first, third, and fifth toes touched with a finger) showed substantial agreement with the monofilament test (5.07) applied over the first, third, and fifth toes and first, third, and fifth metatarsal heads in diabetes patients [31]. However, a study did not detect consistent effects between electrical perceptual threshold and the monofilament test applied over four regions (shoulder, abdomen, and lower limb) in healthy adults (21–68 years) [30], consistent with our findings.

## 5. Conclusions

Our findings on the methodological quality of somatosensory assessments in healthy young adults present a mixed picture. The 128 Hz Rydel-Seiffer tuning fork, applied to the first metatarsal phalangeal joint, malleolus medialis, malleolus lateralis, patella, and anterior superior iliac spine, is a reliable tool for assessing deep somatosensitivity. The commercially available twenty Semmes–Weinstein monofilaments, ranging from 1.65 to 6.65, are reliable for evaluating superficial somatosensitivity of the feet and ankles, but not of the legs, thighs, and trunk. The absence of parallel test reliability between the tuning fork and the monofilament test empirically confirms the theoretical distinction between deep and superficial somatosensation.

## 6. Strengths and Limitations

Our study addresses a gap in the evidence in the field of somatosensory research and supports the development of reliable assessments. The generalisability of our findings is limited by the narrow population studied (limited to healthy young adults) and the relatively small sample size.

## Figures and Tables

**Figure 1 bioengineering-12-01265-f001:**
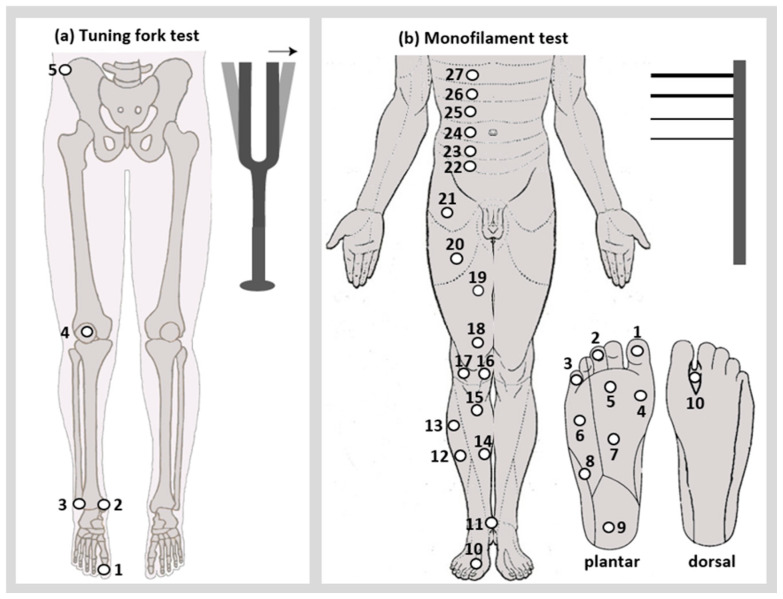
(**a**) The tuning fork test was applied to five different positions of each hemibody; (**b**) the monofilament test was applied to 27 different positions of each hemibody.

**Table 1 bioengineering-12-01265-t001:** Means, SD, intra- and inter-rater reliability (ICC) of the tuning fork test.

	Test Rater 1 (Means and SD)	Retest Rater 1 (Means and SD)	Test Rater 2 (Means and SD)	Intra-Rater Reliability (ICC)	Inter-Rater Reliability (ICC)
Total	47.9 ± 9.5	48.4 ± 11.0	46.8 ± 8.4	0.714	0.601
Right hemibody	23.8 ± 5.0	24.4 ± 5.8	23.4 ± 4.2	0.723	0.644
First metatarsal–phalangeal joint	5.1 ± 1.1	5.2 ± 1.2	5.0 ± 1.1	0.658	0.666
Malleolus medialis	5.1 ± 1.0	5.2 ± 1.2	5.0 ± 0.8	0.669	0.545
Malleolus lateralis	5.0 ± 1.1	5.2 ± 1.3	4.9 ± 1.0	0.687	0.497
Patella	4.1 ± 1.2	4.2 ± 1.4	4.1 ± 1.0	0.671	0.671
Anterior superior iliac spine	4.4 ± 1.1	4.5 ± 1.3	4.3 ± 1.0	0.690	0.558
Left hemibody	24.1 ± 4.7	24.3 ± 5.4	23.4 ± 4.5	0.654	0.522
First metatarsal–phalangeal joint	5.3 ± 1.0	5.1 ± 1.0	5.0 ± 1.0	0.515	0.538
Malleolus medialis	5.2 ± 1.0	5.4 ± 1.1	5.1 ± 0.9	0.500	0.504
Malleolus lateralis	5.0 ± 1.0	5.1 ± 1.1	4.9 ± 0.9	0.644	0.425
Patella	4.2 ± 1.2	4.3 ± 1.2	4.1 ± 1.1	0.539	0.487
Anterior superior iliac spine	4.4 ± 1.1	4.4 ± 1.4	4.3 ± 1.1	0.645	0.457

**Notes: 0.9 ≤ ICC = excellent; 0.9 > ICC ≥ 0.75 = good**; 0.75 > ICC ≥ 0.5 = moderate; ICC < 0.5 = poor; ICC = intraclass correlation coefficient; SD = standard deviation.

**Table 2 bioengineering-12-01265-t002:** Means, SD, intra- and inter-rater reliability (ICC) of the monofilament test.

	Test Rater 1 (Means and SD)	Retest Rater 1 (Means and SD)	Test Rater 2 (Means and SD)	Intra-Rater Reliability (ICC)	Inter-Rater Reliability (ICC)
Total	156.42 ± 19.12	152.97 ± 13.82	154.99 ± 19.10	0.347	0.374
Right hemibody	78.21 ± 8.97	77.06 ± 7.16	78.03 ± 9.76	0.355	0.185
Foot, ankle	37.84 ± 3.38	37.75 ± 3.33	38.17 ± 3.47	**0.801**	**0.785**
P1	3.52 ± 0.35	3.49 ± 0.40	3.57 ± 0.31	0.655	0.659
P2	3.44 ± 0.41	3.38 ± 0.42	3.43 ± 0.39	0.690	0.719
P3	3.44 ± 0.40	3.39 ± 0.42	3.41 ± 0.45	**0.764**	0.714
P4	3.46 ± 0.38	3.46 ± 0.38	3.55 ± 0.33	0.724	0.745
P5	3.41 ± 0.37	3.43 ± 0.40	3.47 ± 0.41	0.656	0.599
P6	3.48 ± 0.31	3.48 ± 0.32	3.52 ± 0.36	0.605	0.444
P7	3.20 ± 0.54	3.23 ± 0.46	3.20 ± 0.51	0.638	0.617
P8	3.41 ± 0.37	3.38 ± 0.43	3.50 ± 0.34	0.672	0.665
P9	3.67 ± 0.32	3.68 ± 0.31	3.69 ± 0.36	0.508	0.583
P10	3.15 ± 0.63	3.17 ± 0.52	3.13 ± 0.61	0.347	0.333
P11	3.65 ± 0.56	3.67 ± 0.52	3.70 ± 0.56	0.557	0.714
Leg, thigh	29.17 ± 5.65	28.45 ± 5.43	28.05 ± 5.92	0.021	−0.193
P12	3.50 ± 0.59	3.54 ± 0.49	3.47 ± 0.64	0.076	−0.036
P13	3.45 ± 0.62	3.41 ± 0.59	3.39 ± 0.62	0.042	−0.283
P14	3.36 ± 0.62	3.37 ± 0.65	3.30 ± 0.67	0.120	−0.220
P15	3.34 ± 0.66	3.13 ± 0.78	3.16 ± 0.84	−0.055	0.191
P16	2.42 ± 0.78	2.19 ± 0.75	2.37 ± 0.80	−0.277	−0.284
P17	2.61 ± 0.88	2.36 ± 0.78	2.44 ± 0.79	0.145	0.251
P18	2.24 ± 0.75	2.22 ± 0.69	2.21 ± 0.74	0.357	0.159
P19	2.61 ± 0.82	2.63 ± 0.86	2.45 ± 0.80	−0.285	−0.205
P20	2.79 ± 0.86	2.68 ± 0.79	2.51 ± 0.85	0.062	−0.025
P21	2.85 ± 0.91	2.92 ± 0.87	2.75 ± 0.91	0.266	−0.515
Trunk	11.21 ± 2.57	10.86 ± 1.85	11.82 ± 3.60	0.268	0.227
P22	2.10 ± 0.71	1.96 ± 0.56	2.08 ± 0.77	0.144	0.215
P23	1.84 ± 0.51	1.84 ± 0.43	2.04 ± 0.70	0.148	0.091
P24	1.88 ± 0.54	1.73 ± 0.26	1.89 ± 0.57	0.055	0.232
P25	1.77 ± 0.46	1.74 ± 0.34	1.93 ± 0.62	0.182	0.190
P26	1.80 ± 0.39	1.76 ± 0.36	1.91 ± 0.64	0.370	0.286
P27	1.83 ± 0.44	1.81 ± 0.40	1.98 ± 0.69	0.608	0.028
Left hemibody	78.20 ± 10.67	75.91 ± 7.05	76.96 ± 9.73	−0.128	0.358
Foot, ankle	37.55 ± 3.54	37.15 ± 3.40	37.71 ± 3.52	**0.782**	**0.804**
P1	3.47 ± 0.41	3.43 ± 0.36	3.52 ± 0.30	0.716	0.727
P2	3.39 ± 0.42	3.32 ± 0.40	3.39 ± 0.38	0.642	0.454
P3	3.45 ± 0.40	3.42 ± 0.36	3.42 ± 0.38	0.640	**0.786**
P4	3.45 ± 0.39	3.41 ± 0.43	3.47 ± 0.37	0.469	0.655
P5	3.39 ± 0.37	3.41 ± 0.39	3.43 ± 0.38	0.585	0.573
P6	3.42 ± 0.36	3.46 ± 0.35	3.46 ± 0.41	0.694	0.722
P7	3.15 ± 0.55	3.19 ± 0.51	3.25 ± 0.43	0.733	**0.795**
P8	3.32 ± 0.43	3.25 ± 0.44	3.32 ± 0.45	**0.793**	**0.754**
P9	3.68 ± 0.36	3.64 ± 0.32	3.73 ± 0.29	0.433	0.341
P10	3.20 ± 0.55	3.08 ± 0.60	3.11 ± 0.64	0.361	0.392
P11	3.62 ± 0.63	3.52 ± 0.65	3.60 ± 0.71	0.539	0.658
Leg, thigh	29.04 ± 6.52	28.19 ± 5.67	27.97 ± 6.71	0.271	0.076
P12	3.48 ± 0.57	3.52 ± 0.48	3.34 ± 0.72	0.223	−0.110
P13	3.40 ± 0.72	3.37 ± 0.55	3.32 ± 0.79	0.205	0.135
P14	3.36 ± 0.80	3.26 ± 0.73	3.33 ± 0.77	0.283	0.276
P15	3.16 ± 0.90	3.05 ± 0.76	3.07 ± 0.88	−0.132	0.161
P16	2.52 ± 0.86	2.21 ± 0.75	2.39 ± 0.77	0.303	0.168
P17	2.64 ± 0.90	2.49 ± 0.83	2.47 ± 0.85	0.077	0.014
P18	2.31 ± 0.76	2.26 ± 0.72	2.21 ± 0.69	0.276	0.384
P19	2.58 ± 0.85	2.54 ± 0.81	2.46 ± 0.87	0.013	−0.052
P20	2.68 ± 0.92	2.65 ± 0.85	2.61 ± 0.89	0.014	−0.162
P21	2.94 ± 0.89	2.84 ± 0.87	2.77 ± 0.91	0.095	−0.228
Trunk	11.61 ± 3.15	10.58 ± 1.08	11.28 ± 2.54	−0.003	0.326
P22	2.07 ± 0.73	1.87 ± 0.47	2.00 ± 0.70	0.162	0.442
P23	1.92 ± 0.59	1.72 ± 0.22	1.90 ± 0.55	−0.033	−0.174
P24	1.92 ± 0.61	1.71 ± 0.21	1.81 ± 0.36	−0.186	−0.015
P25	1.87 ± 0.57	1.73 ± 0.26	1.79 ± 0.41	−0.191	0.297
P26	1.88 ± 0.59	1.74 ± 0.29	1.81 ± 0.41	0.416	0.393
P27	1.94 ± 0.62	1.80 ± 0.36	1.97 ± 0.65	0.403	0.231

**Notes: 0.9 ≤ ICC = excellent; 0.9 > ICC ≥ 0.75 = good**; 0.75 > ICC ≥ 0.5 = moderate; ICC < 0.5 = poor; ICC = intraclass correlation coefficient; SD = standard deviation.

**Table 3 bioengineering-12-01265-t003:** Internal consistency (Cronbach’s α) of the tuning fork test.

		Total	Right Hemibody	First Metatarsal–Phalangeal Joint	Malleolus Medialis	Malleolus Lateralis	Patella	Anterior Superior Iliac Spine	Left Hemibody	First Metatarsal–Phalangeal Joint	Malleolus Medialis	Malleolus Lateralis	Patella
Right hemibody		**0.895**											
First metatarsal–phalangeal joint		0.322	0.528										
Malleolus medialis		0.328	0.539	**0.921**									
Malleolus lateralis		0.356	0.580	**0.913**	**0.929**								
Patella		0.373	0.585	**0.832**	**0.897**	**0.905**							
Anterior superior iliac spine		0.321	0.527	0.770	**0.861**	**0.864**	**0.840**						
Left hemibody		**0.872**	**0.948**	0.512	0.522	0.552	0.593	0.508					
First metatarsal–phalangeal joint		0.276	0.411	**0.821**	0.790	0.728	0.756	0.673	0.504				
Malleolus medialis		0.301	0.461	**0.851**	**0.875**	**0.863**	**0.851**	0.785	0.529	**0.833**			
Malleolus lateralis		0.318	0.484	**0.855**	**0.890**	**0.866**	**0.867**	**0.804**	0.548	**0.806**	**0.941**		
Patella		0.360	0.547	**0.815**	**0.861**	**0.884**	**0.934**	**0.808**	0.597	0.760	**0.853**	**0.839**	
Anterior superior iliac spine		0.339	0.518	0.787	**0.854**	**0.853**	**0.872**	**0.905**	0.573	0.767	**0.826**	**0.873**	**0.876**

**Notes: 0.9 ≤ α = excellent; 0.9 > α ≥ 0.8 = good**; 0.8 > α ≥ 0.7 = acceptable; 0.7 > α ≥ 0.6 = questionable; 0.6 > α ≥ 0.5 = poor; α < 0.5 = unacceptable.

**Table 4 bioengineering-12-01265-t004:** Internal consistency (Cronbach’s α) of the monofilament test.

			Right Hemibody	Foot, Ankle	Leg, Thigh	Trunk	Left Hemibody	Foot, Ankle	Leg, Thigh
		Total							
Right hemibody		**0.854**							
Foot, ankle		0.351	0.586						
Leg, thigh		0.642	**0.896**	0.433					
Trunk		0.292	0.532	0.297	0.564				
Left hemibody		**0.909**	**0.938**	0.506	0.799	0.421			
Foot, ankle		0.379	0.587	**0.951**	0.450	0.342	0.565		
Leg, thigh		0.711	**0.886**	0.454	**0.935**	0.497	**0.899**	0.504	
Trunk		0.371	0.564	0.372	0.497	**0.840**	0.577	0.434	0.631

**Notes: 0.9 ≤ α = excellent; 0.9 > α ≥ 0.8 = good;** 0.8 > α ≥ 0.7 = acceptable; 0.7 > α ≥ 0.6 = questionable; 0.6 > α ≥ 0.5 = poor; α < 0.5 = unacceptable.

**Table 5 bioengineering-12-01265-t005:** Internal consistency (Cronbach’s α) of the monofilament test (foot and ankle subscale).

Right foot and ankle		**Right Foot and Ankle**											**Left Foot and Ankle**									
		**P1**	**P2**	**P3**	**P4**	**P5**	**P6**	**P7**	**P8**	**P9**	**P10**	**P11**	**P1**	**P2**	**P3**	**P4**	**P5**	**P6**	**P7**	**P8**	**P9**	**P10**
	P2	0.683																				
	P3	0.698	0.613																			
	P4	0.662	0.696	0.716																		
	P5	0.546	0.674	0.736	0.789																	
	P6	0.395	0.519	0.741	0.649	0.770																
	P7	0.707	0.599	0.735	0.773	0.737	0.577															
	P8	0.658	0.591	0.744	**0.819**	**0.816**	**0.814**	0.785														
	P9	0.510	0.425	0.681	0.508	0.587	0.660	0.479	0.427													
	P10	0.494	0.617	0.537	0.509	0.548	0.441	0.661	0.496	0.457												
	P11	0.625	0.640	0.591	0.642	0.456	0.530	0.746	0.630	0.365	0.652											
Left foot and ankle	P1	0.749	0.740	0.737	0.796	0.773	0.509	0.672	0.567	0.457	0.550	0.577										
	P2	0.612	0.607	0.735	0.715	0.764	0.646	0.599	0.570	0.449	0.462	0.468	**0.822**									
	P3	0.563	0.673	0.674	0.698	0.767	0.599	0.689	0.681	0.348	0.552	0.490	0.700	0.750								
	P4	0.602	0.572	0.637	**0.818**	0.754	0.764	0.650	0.783	0.435	0.552	0.571	0.697	0.682	0.591							
	P5	0.628	0.537	0.649	0.775	**0.858**	**0.813**	0.708	0.796	0.521	0.503	0.559	0.692	0.677	0.708	**0.869**						
	P6	0.661	0.639	0.720	**0.837**	**0.837**	**0.805**	0.725	**0.834**	0.669	0.564	0.550	0.643	0.668	0.737	**0.867**	**0.903**					
	P7	0.545	0.612	0.697	0.634	0.695	0.643	0.793	0.682	0.459	0.704	0.658	0.609	0.640	0.672	0.586	0.673	0.693				
	P8	0.566	0.638	**0.815**	0.771	**0.816**	**0.822**	0.791	**0.834**	0.630	0.548	0.655	0.714	0.764	0.705	0.747	**0.810**	**0.856**	0.789			
	P9	0.477	0.695	0.603	0.511	0.479	0.597	0.465	0.422	0.695	0.456	0.362	0.498	0.569	0.607	0.360	0.517	0.651	0.608	0.654		
	P10	0.479	0.565	0.460	0.489	0.534	0.280	0.515	0.428	0.426	0.781	0.454	0.625	0.510	0.359	0.424	0.410	0.538	0.633	0.587	0.497	
	P11	0.529	0.594	0.482	0.622	0.517	0.555	0.705	0.561	0.373	0.684	**0.871**	0.591	0.506	0.495	0.598	0.610	0.560	0.587	0.664	0.346	0.447

**Notes: 0.9 ≤ α = excellent; 0.9 > α ≥ 0.8 = good**; 0.8 > α ≥ 0.7 = acceptable; 0.7 > α ≥ 0.6 = questionable; 0.6 > α ≥ 0.5 = poor; α < 0.5 = unacceptable.

**Table 6 bioengineering-12-01265-t006:** Internal consistency (Cronbach’s α) of the monofilament test (leg and thigh subscale).

		**Right Leg and Thigh**										**Left Leg and Thigh**								
		**P12**	**P13**	**P14**	**P15**	**P16**	**P17**	**P18**	**P19**	**P20**	**P21**	**P12**	**P13**	**P14**	**P15**	**P16**	**P17**	**P18**	**P19**	**P20**
Right leg and thigh	P13	0.683																		
	P14	0.698	0.613																	
	P15	0.662	0.696	0.716																
	P16	0.546	0.674	0.736	0.789															
	P17	0.395	0.519	0.741	0.649	0.770														
	P18	0.701	0.599	0.735	0.773	0.737	0.577													
	P19	0.658	0.591	0.744	**0.819**	**0.816**	**0.814**	0.785												
	P20	0.510	0.425	0.681	0.508	0.587	0.660	0.479	0.427											
	P21	0.494	0.617	0.537	0.508	0.548	0.441	0.661	0.496	0.457										
Left leg and thigh	P12	0.515	0.639	0.493	0.437	0.497	0.339	0.538	0.405	0.212	0.606									
	P13	0.478	0.589	0.439	0.331	0.503	0.266	0.472	0.347	0.142	0.742	**0.858**								
	P14	0.390	0.531	0.407	0.287	0.466	0.234	0.463	0.308	0.105	0.645	0.829	**0.945**							
	P15	0.459	0.516	0.308	0.378	0.460	0.161	0.532	0.292	0.079	0.690	0.697	**0.890**	**0.884**						
	P16	0.187	0.258	0.163	0.092	0.217	−0.013	0.379	0.071	−0.092	0.559	0.469	0.697	0.711	0.750					
	P17	0.055	0.132	−0.048	−0.109	0.085	−0.079	0.253	−0.007	−0.222	0.370	0.416	0.672	0.680	0.686	**0.912**				
	P18	−0.005	0.123	−0.059	0.099	0.109	−0.108	0.337	0.119	−0.425	0.351	0.331	0.457	0.509	0.526	**0.841**	0.799			
	P19	0.202	0.244	0.092	0.102	0.152	−0.142	0.381	0.032	−0.158	0.438	0.520	0.647	0.672	0.728	**0.887**	**0.853**	**0.835**		
	P20	0.207	0.307	0.123	0.210	0.134	0.013	0.405	0.182	−0.262	0.450	0.540	0.633	0.688	0.732	**0.805**	0.797	0.770	**0.906**	
	P21	0.161	0.253	−0.001	0.084	0.020	−0.090	0.316	−0.045	−0.132	0.331	0.428	0.594	0.630	0.677	**0.800**	**0.824**	0.729	**0.904**	**0.907**

**Notes: 0.9 ≤ α = excellent; 0.9 > α ≥ 0.8 = good**; 0.8 > α ≥ 0.7 = acceptable; 0.7 > α ≥ 0.6 = questionable; 0.6 > α ≥ 0.5 = poor; α < 0.5 = unacceptable.

**Table 7 bioengineering-12-01265-t007:** Internal consistency (Cronbach’s α) of the monofilament test (trunk subscale).

		**Right Trunk**						**Left Trunk**				
		**P22**	**P23**	**P24**	**P25**	**P26**	**P27**	**P22**	**P23**	**P24**	**P25**	**P26**
Right trunk	P23	0.682										
	P24	0.689	**0.925**									
	P25	0.718	**0.913**	**0.904**								
	P26	0.601	**0.831**	0.794	**0.858**							
	P27	0.657	0.677	0.699	0.748	**0.854**						
Left trunk	P22	**0.872**	0.638	0.620	0.671	0.663	0.583					
	P23	0.740	0.758	0.721	0.758	0.700	0.537	**0.856**				
	P24	0.750	0.663	0.682	0.704	0.693	0.614	**0.882**	**0.914**			
	P25	0.552	0.583	0.712	0.665	0.745	0.563	0.695	**0.801**	**0.877**		
	P26	0.590	0.517	0.680	0.631	0.748	**0.802**	0.631	0.639	0.769	**0.880**	
	P27	0.459	0.543	0.682	0.620	0.767	0.761	0.578	0.646	0.772	**0.890**	**0.947**

**Notes: 0.9 ≤ α = excellent; 0.9 > α ≥ 0.8 = good**; 0.8 > α ≥ 0.7 = acceptable; 0.7 > α ≥ 0.6 = questionable; 0.6 > α ≥ 0.5 = poor; α < 0.5 = unacceptable.

**Table 8 bioengineering-12-01265-t008:** Parallel test reliability (ICC) of the monofilament and tuning fork test.

		**Tuning Fork Test**												
		**Total**	**Right Hemibody**	**First Metatarsal-Phalangeal Joint**	**Malleolus Medialis**	**Malleolus Lateralis**	**Patella**	**Anterior Superior Iliac Spine**	**Left Hemibody**	**First Metatarsal-Phalangeal Joint**	**Malleolus Medialis**	**Malleolus Lateralis**	**Patella**	**Anterior Superior Iliac Spine**
Monofilament Test	Total	0.132	0.035	0.022	0.019	−0.009	0.013	−0.008	0.122	0.064	0.022	0.032	0.007	0.011
	Right hemibody	0.056	−0.012	0.035	0.022	−0.045	0.011	−0.041	0.103	0.105	0.007	0.027	−0.002	−0.006
	Foot, ankle	0.250	0.288	0.249	0.207	0.108	0.185	0.144	0.400	0.386	0.231	0.258	0.182	0.102
	Leg and thigh	−0.117	−0.193	0.010	−0.021	−0.130	−0.020	−0.164	−0.065	0.092	−0.060	−0.029	−0.078	−0.038
	Trunk	−0.063	−0.120	−0.143	−0.055	−0.130	−0.136	−0.027	−0.082	−0.019	−0.108	−0.074	−0.025	−0.082
	Left hemibody	0.235	0.103	0.046	0.046	−0.009	0.032	0.004	0.253	0.123	0.064	0.080	0.022	0.039
	Foot, ankle	0.295	0.345	0.275	0.225	0.001	0.190	0.171	0.445	0.370	0.258	0.301	0.185	0.137
	Leg and thigh	0.105	0.010	0.044	0.023	−0.038	0.026	−0.043	0.197	0.149	0.050	0.064	−0.004	0.049
	Trunk	0.039	−0.042	−0.087	0.084	−0.077	−0.036	−0.024	0.157	0.158	0.124	0.158	−0.014	0.039

**Notes: 0.9 ≤ ICC = excellent; 0.9 > ICC ≥ 0.75 = good;** 0.75 > ICC ≥ 0.5 = moderate; ICC < 0.5 = poor; ICC = intraclass correlation coefficient; SD = standard deviation.

## Data Availability

The datasets generated during and/or analysed during the current study are available from the corresponding author upon reasonable request.

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
