# Peer review of "Intra- and Inter-Rater Reliability, Parallel Test Reliability, and Internal Consistency of the Tuning Fork and Monofilament Tests"

_bioengineering, 2025, doi:10.3390/bioengineering12111265_

Round 1
Reviewer 1 Report
Comments and Suggestions for Authors
Minor Comments:
- Which sampling (probable or non-probable, etc.) method was used in the study?
- Statistical tests for hypothesis testing and their assumptions should be specified in the study's statistical analysis in the Materials and Methods section.
- The abstract states the monofilament test was applied to "27 locations of each hemibody," while the methods section (2.3.2) and Figure 1b reference 27 locations total for each hemibody, which is consistent. However, the results tables (Table 2) list items P1-P27, implying 27 points per hemibody, which would mean 54 points total. The text consistently refers to 27 locations per hemibody, so the description of the total number of test points in the abstract and methods is correct, but the numbering (P1-P27) might be slightly misleading without explicitly stating it's per hemibody, though the context (right/left hemibody scores) makes it clear.
- In Table 1, the row for "Left hemi-body" has a typo in the header: "Left hemi-body" is written as "Left hemi-body" in the table, but the text uses "Left hemibody" elsewhere (e.g., "Left hemibody score"). This is a minor inconsistency in hyphenation.
- Table 8's title refers to "Tunning Fork Test," which is a clear typographical error and should be "Tuning Fork Test." This is a minor error in the presentation of data.
- The reference to "RR" and "CS" in the Author Contributions section (Line 311) is not defined in the author list (which lists Veldema, Sasse, Straub, Klemm, von Grönheim, Steingräber). This appears to be an error in the contribution statement, as these initials do not correspond to the listed authors. This is a minor but significant administrative inconsistency.
- The Ethics approval line (307) states "EC no. 2022-043," while the Methods section (105) states "EUB-2023-080." This is a discrepancy in the ethics committee approval number. The text should be consistent. This is a minor but important administrative inconsistency.
- The "Notes" section under Table 8 incorrectly states "0.9 ≤ α= excellent" and uses "α" for ICC values, which is a clear mislabeling. ICC values are reported, so the note should refer to "ICC" and not "α". This is a minor error in the table caption that could cause confusion.
- Which methods are used to model relationships between variables?
Major Comments:
- How was the sample size determined? This information should be explained in the Materials and Methods section.
- The core finding of the study is that the monofilament test demonstrates poor intra- and inter-rater reliability for the leg, thigh, and trunk subscales (ICC values consistently < 0.5, often negative), and poor internal consistency for most items in these regions (Cronbach's α values mostly < 0.5 or questionable). This is a major finding that directly challenges the use of standard commercial monofilament kits for objective evaluation of somatosensation in these body regions. The paper presents robust statistical evidence (ICCs and α values) to support this claim, making it a significant contribution to the field.
- The paper demonstrates a complete lack of parallel test reliability (ICC values mostly < 0.2, often near zero or negative) between the tuning fork test (assessing deep vibration sense) and the monofilament test (assessing superficial touch sense). This finding is crucial as it empirically confirms the theoretical distinction between deep and superficial somatosensation, validating their use as distinct constructs. This is a major contribution, as it provides direct evidence against the common practice of using these tests interchangeably or assuming they measure the same underlying sensory function.
- The internal consistency analysis of the monofilament test reveals a highly complex and inconsistent pattern. While the total score and right/left hemibody scores showed acceptable to good consistency, the subscales (foot/ankle, leg/thigh, trunk) showed highly variable internal consistency, with some items showing excellent consistency and others unacceptable. This suggests the monofilament test, particularly for the leg/thigh and trunk, may not form a coherent, unified scale. The finding that adjacent items on the same hemibody showed stronger consistencies than distant or contralateral items further complicates the interpretation of the test's total score and underscores the non-uniform nature of the data. This is a major insight into the limitations of the monofilament test's scoring structure.
- The study explicitly states that the commercially available 20-monofilament kit (1.65-6.65) is insufficient for objective evaluation of leg, thigh, and trunk regions. This is a major practical conclusion drawn directly from the data showing poor reliability and consistency in these areas. The paper provides strong evidence to support this recommendation, which has direct implications for clinical and research practice.
- The paper's conclusion that the tuning fork test is reliable for deep somatosensation in the lower extremities of healthy young adults is supported by moderate to good intra- and inter-rater reliability (ICCs mostly between 0.5 and 0.7) and good internal consistency (Cronbach's α for total score 0.895). This finding is consistent with some existing literature cited and provides a clear, positive assessment of the tuning fork's reliability for its intended purpose in this population, which is a valuable contribution.
Author Response
Dear reviewer, thank you very much for the time taken to review our manuscript. We performed the revision in line with your comments.

Reviewer 2 Report
Comments and Suggestions for Authors
The manuscript is clearly written, logically structured, and requires no major revisions. Figures, tables, and references are appropriate and informative.
In my opinion, this manuscript meets the standards for publication and can be accepted as submitted.
Author Response
Dear reviewer, thank you very much for the positive evaluation of our manuscript.
Reviewer 3 Report
Comments and Suggestions for Authors
In this study, somatosensory sensitivity was assessed in a statistical sample involving different age groups using a tuning fork and a monofilament test. The Semmes-Weinstein monofilament test generally shows high intra- and inter-rater reliability, while the tuning fork test has lower reliability, especially with inter-rater agreement. The experiments were carefully evaluated and verified by complementary tests. The Authors found that using a tuning fork allows for reliable assessment of deep somatosensory function in the lower extremities. While the use of monofilaments was sufficient for examining superficial somatosensory sensitivity of the feet and ankle joints, it was insufficient for objective assessment of the lower leg, thigh, and torso areas.
Some recommendations
- The Authors conduct a comparative analysis of the two tests, but it should be taken into account that the indicated methodologies evaluate different sensory modalities and the assessment of complementary reliability should be carried out with caution.
- The presence of massive, continuous tables of results spanning several pages makes it somewhat difficult to understand the information. Perhaps it would be better to present the key results in the form of graphs (figures) with confidence intervals.
- The authors' conclusion that monofilaments are insufficient for objective assessment of the condition of the legs, thighs, and torso generally corresponds to generally accepted clinical practice and the results of earlier studies.
- The reliability of tests using monofilaments can vary depending on the manufacturer, the number of uses, the experience of the tester, etc., which limits their objective application in different areas of the body without strictly standardized protocols.
Author Response

(The authors gave the same response as above.)

Round 2
Reviewer 1 Report
Comments and Suggestions for Authors
Accept in present form